# Resuscitation and Forensic Factors Influencing Outcome in Penetrating Cardiac Injury

**DOI:** 10.3390/diagnostics14131406

**Published:** 2024-07-01

**Authors:** Astrid Aumaitre, Clémence Delteil, Lucile Tuchtan, Marie-Dominique Piercecchi-Marti, Marc Gainnier, Julien Carvelli, Salah Boussen, Nicolas Bruder, Fouzia Heireche, Thibault Florant, Françoise Gaillat, David Lagier, Alizée Porto, Lionel Velly, Pierre Simeone

**Affiliations:** 1Service de Médecine Légale, Assistance Publique-Hôpitaux de Marseille, 13005 Marseille, France; astrid.aumaitre@ap-hm.fr (A.A.); lucile.tuchtan@ap-hm.fr (L.T.); marie-dominique.piercecchi@ap-hm.fr (M.-D.P.-M.); 2Faculté de Médecine Secteur Nord 51, Boulevard Pierre Dramard, ADES, Aix Marseille University, 13344 Marseille, CEDEX 15, France; 3Réanimation des Urgences, Assistance Publique-Hôpitaux de Marseille, CHU La Timone, 13385 Marseille, France; marc.gainnier@ap-hm.fr (M.G.); julien.carvelli@ap-hm.fr (J.C.); 4Département d’Anesthésie-Réanimation, CHU Timone, Aix Marseille University, 13005 Marseille, France; salah.boussen@ap-hm.fr (S.B.); nicolas.bruder@ap-hm.fr (N.B.); lionel.velly@ap-hm.fr (L.V.); pierre.simeone@ap-hm.fr (P.S.); 5Faculté de Médecine Secteur Nord 51, Boulevard Pierre Dramard, IFSTTAR, LBA UMR_T 24, Aix Marseille University, 13344 Marseille, CEDEX 15, France; 6SAMU13, Pôle RUSH, CHU La Timone, AP-HM, 13005 Marseille, France; fouzia.heireche@ap-hm.fr; 7Department of Public Health, University Hospital of Marseille, 13015 Marseille, France; thibault.florant@ap-hm.fr; 8Département d’Anesthésie et Réanimation (SAR 2), CHU La Timone, Assistance Publique des Hôpitaux de Marseille, 13015 Marseille, France; francoise.gaillat@ap-hm.fr (F.G.); david.lagier@ap-hm.fr (D.L.); 9Département de Chirurgie Cardiaque, CHU Timone, 13005 Marseille, France; alizee.porto@ap-hm.fr; 10Institut des Neurosciences de la Timone, CNRS, Aix Marseille University, 13005 Marseille, France

**Keywords:** emergency medicine, penetrating cardiac injury, forensic, sharp object

## Abstract

Background: Cardiac injury caused by a sharp object is a medical and surgical therapeutic challenge. Mortality risk factors have been identified but there are major discrepancies in the literature. The aim of this study was to analyse the management of victims of penetrating cardiac injuries before and after admission to hospital and the anatomical characteristics of these injuries in order to facilitate diagnosis of the most critical patients. Methods: To carry out this study, we conducted a retrospective analytical study with epidemiological data on victims of penetrating cardiac injuries. We included two types of patients, with those who underwent autopsy in our institution after death from sharp injury to the heart or great vessels and those who survived with treatment in the emergency department or intensive care unit between January 2015 and February 2022. Results: We included 30 autopsied patients and 12 survivors aged between 18 and 73 years. Higher mortality was associated with prehospital or in-hospital cardiorespiratory arrest (OR = 4, CI [1.71–9.35]), preoperative mechanical ventilation (OR = 10, CI [1.53–65.41]), preoperative catecholamines (OR = 7, CI [1.12–6.29]), preoperative and perioperative adrenaline (OR = 13, CI [1.98–85.46] and [1.98–85.46]), penetrating cardiac injury (OR = 14, CI [2.10–93.22]), multiple cardiac injuries (OR = 1.5, CI [1.05–2.22]) and an Organ Injury Scaling of the American Association for the Surgery of Trauma (AAST-OIS) score of 5 (OR = 2.9, CI [1.04–8.54]; *p* = 0.0329) with an AUC-ROC curve value of 0.708 (CI [0.543–0.841]). Conclusions: This study identified risk mortality factors in penetrating cardiac injury patients. These findings can help improve the diagnosis and management of these patients. The AAST-OIS score may be a good tool to diagnose critical patients.

## 1. Introduction

Thoracic injuries caused by sharp objects, particularly when located within the cardiac box, carry a high risk of cardiac injury [1]. The management of patients with penetrating intracardiac injuries is a critical and complex medical and surgical challenge. The outcome of these patients largely depends on the promptness and effectiveness of the initial treatment. Rapid and accurate diagnosis, followed by timely and appropriate interventions, is essential to improve the survival and long-term outcomes of patients with penetrating intracardiac injuries. With advances in diagnostic methods and modernisation of resuscitation and surgical techniques, an increasing number of patients reach the hospital and even the operating room alive [2,3]. Nevertheless, mortality remains high, with a survival rate of 3 to 84% in the 10 largest studies on the subject published between 1968 and 2000 [4].

Mortality risk factors after penetrating cardiac injury are difficult to determine and to interpret because most series are small, the confounding factors analysed are numerous, and rigorous statistical methodology is often lacking [4]. There have nevertheless been attempts to identify factors that influence the mortality rate of these injuries. The literature is very heterogeneous and sometimes conflicting. In particular, the following have been described as mortality risk factors: cardiopulmonary resuscitation performed at the scene [5,6], severe hypotension [3,5,6,7,8,9,10,11], initial Glasgow score ≤8 [3,6,9,10], low haemoglobin [12], left ventricular [5] or multichamber injuries [5,8], or associated internal and/or vascular injuries [3,9,13,14]. Most studies report that gunshot wounds [3,6,8,15,16] and poor physiological condition before hospital arrival [3,12,13,15] are factors of poor prognosis. Prognostic factors need to be identified in order to guide the emergency medical team and to establish optimal medical and surgical therapeutic indications. 

The objective was to analyse the prehospital and in-hospital management of patients with penetrating intracardiac injuries, as well as the post hoc anatomic forensic characteristics of these injuries, in an attempt to identify mortality risk factors within the care setting and diagnose critical patients requiring urgent care.

## 2. Materials and Methods

### 2.1. Study Design and Setting

This two-centre, retrospective, and descriptive study was carried out in two hospitals of the “Assistance Publique des Hopitaux de Marseille” (APHM), the Hôpital Nord and the Hôpital de la Timone. We included patients who died and who underwent autopsy in the forensic department following injury to the heart or the large cardiac vessels caused by a sharp object between January 2015 and February 2022. We also included adult patients who were treated for such injuries in the emergencies or intensive care units of the two hospitals during the same period. Exclusion criteria were patient age under 18 at the time of the event, potentially fatal extracardiac injuries, advanced post-mortem changes of the body when discovered (cutaneous and internal changes making it impossible to reliably identify the wound pathway or pathways), insufficient clinical and/or forensic data, and the family’s refusal to participate. 

Patients who died were included after a search for eligible patients in the annual databases of the forensic department of Marseille that record all autopsies carried out. Research was filtered using the same 35 national social security codes (Classification Commune des Actes Médicaux, CCAM) for traumatic thoracic and cardiac injuries and injuries of the large thoracic and cardiac vessels (Appendix A) and by a free-text search for the terms “cardiac box” and “cardiac area injury”. The autopsy reports and pathology expert reports were extracted from the Thanatos software of the forensic department. Eligible surviving patients were sought by the medical informatics department of our institution. 

### 2.2. Data Collection

Data were collected from medical records, autopsy reports, and post-mortem expert reports (Appendix A). Using the models of Asensio et al. [15] and Kaljusto et al. [17], patients were classified in 3 survival categories according to signs of life at different times during their management: -Group A, patients who survived for a few minutes (death before the arrival of emergency services or before arrival at the hospital).-Group B, patients who survived for several minutes to several hours (death on arrival at the hospital, in the operating room, or during in-hospital management).-Group C, patients with nonfatal initial injuries (survivors).

The Injury Severity Score (ISS) was estimated twice: once at patients’ initial management or on arrival at the hospital as appropriate (according to the initial description of injuries) and, secondly, after injury assessment in the operating room and/or at autopsy for deceased patients (allowing evaluation of internal injuries). The Revised Trauma Score (RTS) was established from patients’ clinical and laboratory data on arrival at the hospital. “Penetrating injury” described an injury passing through a wall, such as a vascular injury penetrating the wall and rupturing the intima, or a myocardial injury crossing the epicardium, myocardium, and endocardium and entering a heart chamber. “Injury system” described the overall internal injuries caused solely by the wounding agent along its trajectory. An injury system may thus involve several anatomical structures. “Fatal injury system” indicated the injury system resulting in death. A single cardiac injury corresponds to injury of a single anatomical entity (for example, the right ventricular wall or the base of the aorta). Injuries caused by the resuscitation manoeuvres were excluded. Data were collected on a single secure anonymised Microsoft Excel^®^ spreadsheet. 

### 2.3. Data Analysis 

Data were tested to determine their normal distribution (Shapiro–Wilk test). They are given as means and standard deviations for continuous or median variables and as interquartile intervals for discrete variables, depending on their distribution. Categorical variables are given as numbers (*n*) and percentages (%) (total numbers may vary due to missing data). Groups were compared according to outcome using Fisher’s exact test or the Student t test depending on their distribution. Pearson’s correlation squared (R2) evaluated the correlation between continuous variables. Associations between the variables studied and mortality were sought by univariate analysis followed by a receiver operating characteristic (ROC) and calculation of its area under the curve (AUC). The area under the curve evaluates the capacity of the score to predict mortality. All statistical analysis was carried out using JMP version 13 software. A *p* value <0.05 was considered significant. 

### 2.4. Ethics Committee Approval 

All data extracted from the forensic reports and the medical records of Assistance Publique-Hôpitaux de Marseille were anonymised. This study was approved by the ethics committee of Aix-Marseille University (n° 2022-05-12-016), approved by the Marseille public hospitals (under the AP-HM Health Data Access Portal number PADS B9EUBB), and complied with data protection regulations (Règlement Général sur la Protection des Données (RGPD), dossier n° 2022-66). 

## 3. Results

### 3.1. Population 

In total, 30 autopsied patients and 12 survivors were included (Figure 1). The majority were men (sex ratio men/women 39/3). Age at time of injury was 24.5 years for survivors and 31.5 years for autopsied patients. The demographic characteristics of the general population are shown in Table 1. Injuries were the result of assault in 39 patients (95.1%) and suicide in 2 (4.9%). During prehospital management by the emergency services, 83.3% (*n* = 35) of patients showed signs of life and resuscitation was attempted in 83.3% (*n* = 35). Median heart rate was 120 (110–140) beats per minute (bpm) and 60.8% of patients were spontaneously ventilating. Median Glasgow score was 14 (3–15). Salvage thoracotomy was performed in four patients, of whom two then underwent bedside sternotomy, and one patient underwent thoracotomy immediately afterwards in the operating room. 

### 3.2. Patients Who Survived for Several Minutes to Several Hours and Survivors (Group B and C)

The characteristics of the population and injury subgroups are shown in Table 1. During prehospital management, group C patients had more limb movement than group B (100% vs. 25%; *p* = 0.0011) and spontaneous ventilation (100% vs. 46%; *p* = 0.0071) with a higher median Glasgow score (15 vs. 3; *p* = 0.0038). Prior to admission, group C required less intubation (20% vs. 67%; *p* = 0.0361), catecholamines (18% vs. 73%; *p* = 0.0300), and adrenaline (9% vs. 64%; *p* = 0.0237) than group B patients and presented less cardiorespiratory arrest at presentation (0% vs. 67%; *p* = 0.0013). On reaching the hospital, group C patients had a significantly higher median Glasgow score (15 vs. 3; *p* = 0.0024). Group B were more often in shock (16% vs. 100%; *p* = 0.0128) and had more cardiorespiratory arrests (0% vs. 75%; *p* = 0.0003) than group C patients.

In the operating room, group B patients required more catecholamines (100% vs. 50%; *p* = 0.0152), particularly adrenaline (90% vs. 0%; *p* < 0.0001) than group C patients. Group B patients had more multiple cardiac injuries than group C (58% vs. 16%; *p* = 0.0133). Regarding gravity scores, group B patients had a higher median ISS score (75 vs. 25; *p* < 0.0001), lower median RTS score (1.82 vs. 7.84; *p* = 0.0010), and higher median AAST-OIS score (5 vs. 4; *p* = 0.0353) than group C.

### 3.3. Autopsied Patients (Group A and B Versus Group C)

Specific variables were analysed in autopsied patients (Appendix A). Cause of death was identified as haemorrhagic shock in 76.7% (*n* = 23) of patients, multiorgan failure in 13.3% (*n* = 4), and tamponade in 10% (*n* = 3), whereas more autopsied patients had multiple injuries than survivors (53% vs. 16.6%; *p* = 0.0415). Autopsied patients had more multiple cardiac injuries than survivors, with a median of 2 vs. 1.2 for the survivors (*p* = 0.0162). Location of cardiac injury did not significantly differ between the two groups. Injuries were penetrating in 93.3% (*n* = 28) of autopsied patients and in 50% (*n* = 5) of survivors (*p* = 0.0062). More autopsied patients had associated injuries in areas of attack and defence (traumatic lesions in defence zones (hands and forearms) or grip zones (wrists and shoulders)) (66% vs. 50%; *p* = 0.0195). 

### 3.4. Risks Factors for Mortality

Regarding gravity scores, median AAST-OIS score was higher in autopsied patients (5 vs. 4; *p* = 0.0343). Pre-existing cardiac condition, mode of injury, or median initial ISS score (25 vs. 26; *p* = 0.9365) did not significantly differ between survivors and autopsied patients. Median postoperative/post-autopsy ISS score was lower for survivors than autopsied patients (25 vs. 75; *p* < 0.0001).

Univariate analysis by logistic regression was performed to explore the association between probability of death and the variables showing a statistically significant difference between autopsied patients and survivors (Table 2). The factors associated with higher mortality were in-hospital cardiorespiratory arrest (OR = 4, CI [1.71–9.35]; *p* = 0.0013), preoperative mechanical ventilation (OR = 10, CI [1.53–65.41]; *p* = 0.0003), preoperative catecholamines (OR = 7, CI [1.12–6.29]; *p* < 0.0001), preoperative and perioperative adrenaline (OR = 13, CI [1.98–85.46]; *p* < 0.0001 and OR = 2.7, CI [1.98–85.46]; *p* < 0.0001, respectively), penetrating cardiac injury (OR = 14, CI [2.10–93.22]; *p* = 0.0064), multiple cardiac injuries (OR = 1.5, CI [1.05–2.22]; *p* = 0.0415), and an AAST-OIS score of 5 (OR = 2.9, CI [1.04–8.54]; *p* = 0.0329) with an area under the receiver operating characteristic curve (ROC-AUC) value of 0.708, CI [0.543–0.841] with a specificity of 80.0%, a sensibility of 56.6%, a positive likelihood ratio of 2.83, and a negative likelihood ratio of 0.54 (Figure 2). 

## 4. Discussion

Our study of factors influencing outcome in patients with penetrating cardiac injuries revealed that prehospital parameters differed significantly between autopsied patients and survivors. Autopsied patients had less limb movement and spontaneous ventilation, a lower Glasgow score, and more frequent chest tube placement, cardiorespiratory arrest, and catecholamine administration. A larger proportion of autopsied patients had multiple cardiac injuries, more penetrating injuries, and more injuries in areas of attack and defence, with a higher AAST-OIS score and a higher post-autopsy ISS score. Patients who died during management (group B) had a poorer physiological condition (less limb movement and spontaneous ventilation, lower median Glasgow score, and more frequent state of shock on reaching hospital, prehospital intubation, and catecholamine administration) than survivors (group C). 

In our study, nearly half the patients died at the scene or were dead on reaching the hospital. Asensio et al. [6] reported that 10.5% of patients were pronounced dead at the scene by the emergency medical team and that only 54% of patients reached the operating room alive. In the cohort of 240 patients of Isaza-Restrepo et al. [16], 3% were dead on arrival at the hospital. However, as in other published series, poor physiological condition on admission was a risk factor for short-term mortality [3,5,11,15]. Salvage thoracotomy, generally carried out in the emergency room, is still classically reserved for the most critical patients [18]. However, unlike Bamous et al. [13], we did not observe a significant difference in use of thoracotomy between operated patients who died and survivors. 

Injury location did not significantly differ between survivors and patients who died. In the latter, injury was preferentially located in the left ventricle (38.1%), whereas, in survivors, the right ventricle was involved (11.9%), in contrast to the finding of Bamous et al. [13]. Occupying the greater part of the anterior aspect of the heart, the right ventricle has been described as being the chamber that is the most vulnerable to penetrating injury and is more frequently involved [3,4,5,8,16]. Some studies found a higher mortality risk for left ventricular lesions [5] and others for the right ventricle [13]. Because of its preponderant role in maintaining cardiac output, left ventricular injuries have often been considered as more lethal [15], but opinion is not unanimous. 

In survivors, cardiac injuries generally involved a single heart chamber, whereas, in autopsied patients, injuries involved more than one chamber. The presence of associated extracardiac injuries has sometimes been considered to carry a poor prognosis [3,9,13,14], but this was not confirmed in our study. 

Nevertheless, injuries in areas of attack and defence increased the risk of death, doubtless reflecting more violent assault. The penetrating nature of injury has not been studied in the literature as a potential mortality risk factor but appears to be of fundamental importance. In the present study, penetrating injuries significantly predominated in patients who died, particularly in patients who did not reach hospital alive. Whereas a tangential injury does not perforate a heart cavity, a penetrating injury breaches the integrity of the myocardium, potentially causing massive haemorrhage and compromising cardiac haemodynamic. This accounts for the more lethal nature of these injuries, which increased the risk of death in our study. 

Numerous studies have aimed to define the clinical, paraclinical, or anatomical mortality risk factors of multiple penetrating cardiac injuries. Clinically, numerous authors agree that hypotension [3,5,7,8,9,10,11], cardiac arrest [11,19], Glasgow score <8 [3,9,10,15], absence of limb movement, and of spontaneous ventilation [6] indicate poor prognosis. Similarly, we observed that absence of limb movement and of spontaneous ventilation at initial management and cardiorespiratory arrest at the scene and in hospital were mortality risk factors. 

Regarding medical management of these patients, some studies have found, as we did, that cardiopulmonary resuscitation at the scene [5,15,17] and salvage thoracotomy were mortality risk factors [15]. Preoperative mechanical ventilation was also found to be a factor of increased mortality. In our study, pre- and perioperative administration of adrenaline were associated with a higher mortality risk, each factor with OR = 13. They reflect the precarious haemodynamic status of these patients and the gravity of their clinical condition. As these factors have not been examined in the literature, further study would be required to confirm their status or impact as mortality risk factors. It is also interesting to note that the time to care was also longer in the survivor group. However, it is difficult to conclude whether this shows the true superiority of the stay-and-play strategy or whether the most severe patients were deliberately transferred to hospital more quickly.

The severity of anaemia has also been described as a factor that affects mortality [12]. We did not observe a significant difference in haemoglobin levels at various timepoints during management or in transfusion requirement and number of units. Mortality risk was, however, increased by chest tube placement during prehospital management. Massive haemorrhage, related in particular to haemothorax or haemopericardium, is one of the principal causes of death from penetrating cardiac injury by a sharp object [4,20] and it was the main cause of death in our study. The prognostic significance of tamponade is still debated. It has been described as a protective factor [5,21,22,23], a mortality risk factor [13], or as having no particular impact [24,25]. We found no significant difference in presence or absence of tamponade between autopsied patients and survivors. More recent studies [4,26] tend to show that tamponade may have both a protective and a harmful effect. If blood collects rapidly in the pericardium, exceeding its capacity for compliance, intrapericardial pressure increases and exceeds right ventricular pressure and the filling capacity of the left ventricle. This explains why gradual effusion is better tolerated than effusion that builds up rapidly. The protective effect of tamponade is due initially to limitation of blood depletion and the harmful effect appears when the capacities for tolerance and adaptation of the pericardium and the heart are exceeded and cardiorespiratory arrest could follow.

The AAST-OIS score [27] has frequently been used in the literature to attempt to correlate mortality with the severity of cardiac injuries. Isaza-Restrepo et al. [16] reported a 22.6% mortality rate for grade 5 injuries and 16.7% for grade 4 injuries. These findings are in agreement with those of Bamous et al. [13] and Asensio et al. [15]. We too found that autopsied patients had a significantly higher AAST-OIS score than survivors, with a score of 5 carrying a higher mortality risk. This score seems an important predictor of patient outcome, but its use requires full injury evaluation (through complementary postsurgical investigations or autopsy) to assess the absence or presence of tamponade and the extent of injury. Finally, such a score may be useful, essentially in order to diagnose patients at greatest risk of death [10,27,28,29].

This study has several limitations. Firstly, it is a retrospective study whose recruitment is based on registries but whose final representativeness remains limited. Secondly, as with all studies of this type, there are undoubtedly several confounding factors at the time of treatment for which we have no data and therefore cannot adjust for, which could bias the results. Thirdly, the nosological entity of cardiac wounds remains difficult to characterise, particularly in the acute phase, and represents a real challenge for diagnostic categorisation. Instead, the depth of the wound was not a factor considered. Fourth, some variables may vary during management, introducing a random element into the data.

## 5. Conclusions

In this retrospective study of patients with penetrating cardiac injury, comparison of survivors and patients who died showed that those who died had a poorer physiological state before and after hospital admission and that their resuscitation requirements were greater. Although the cardiac location of injury did not significantly differ between groups, its penetrating nature, which receives little mention in the literature, emerged as a major factor of increased mortality, together with associated active or passive defence injuries. The different predictive and anatomical scores showed that mortality risk was higher with an AAST-OIS score of 5. This score appears to be a good tool to diagnose severe patients with higher risk of death.

## Figures and Tables

**Figure 1 diagnostics-14-01406-f001:**
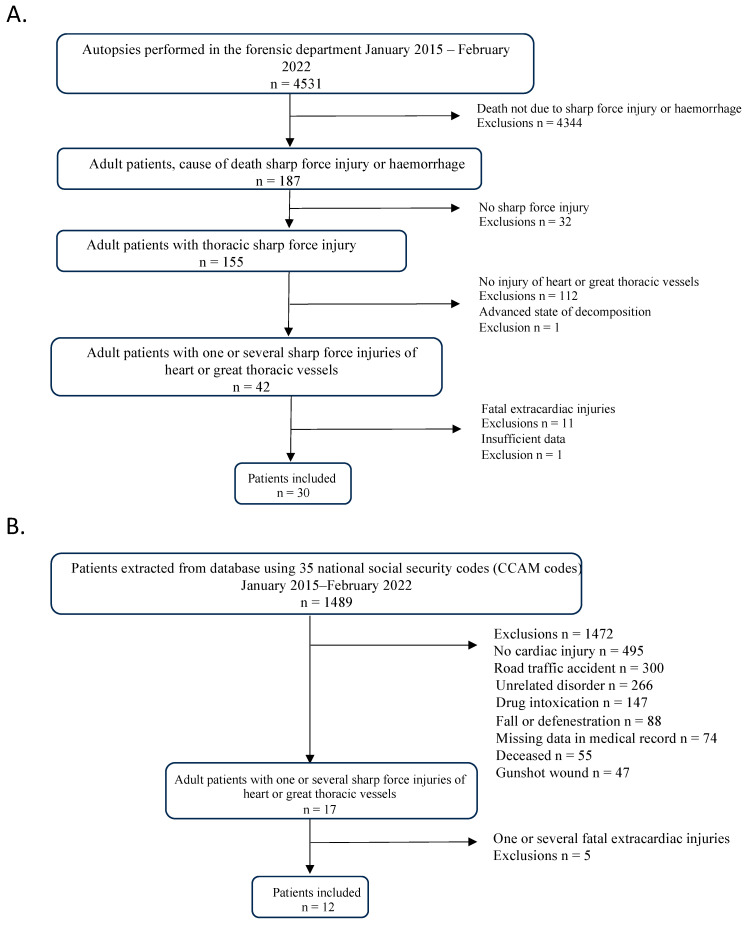
Flowchart of autopsied patients (**A**) and survivors (**B**).

**Figure 2 diagnostics-14-01406-f002:**
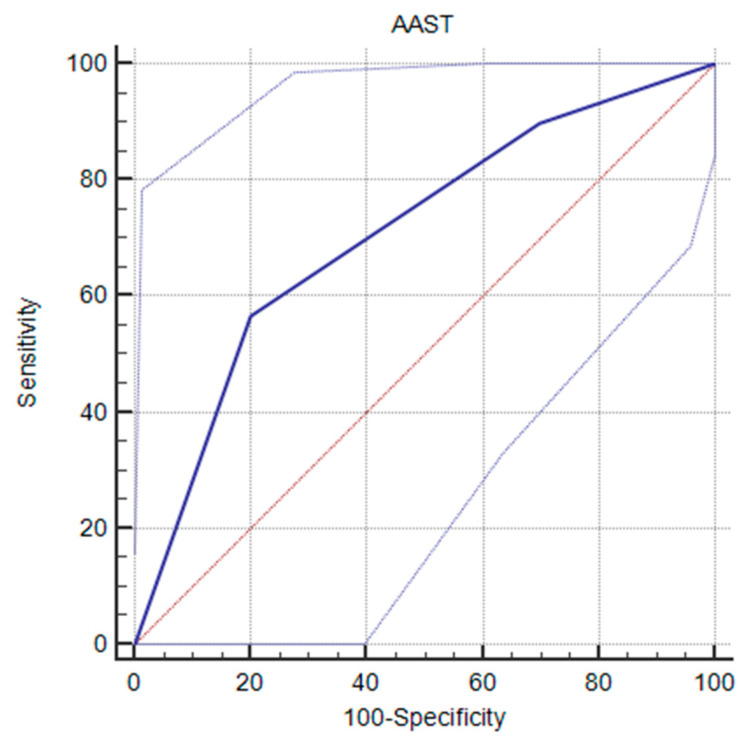
Receiver operating characteristic (ROC) curve of AAST score by probability of death (blue line), the red line is the AUC line at 0.5, and the two gray lines are the confidence intervals.

**Table 1 diagnostics-14-01406-t001:** Characteristics and comparison of subgroups. Group A, survival for a few minutes. Group B, survival for several minutes to several hours. Group C, nonfatal injuries (survivor).

Variables	All Patients	Group A	Group B	Group C	Between-Group Comparison *p*-Value		
	*N* = 42	*N* = 18	*N* = 12	*N* = 12	*p*-Value A–B	*p*-Value A–C	*p*-Value B–C
**Age, years (range)**	31 (24–39.25)	34 (26–59)	31.5 (25–34)	24.5 (20–36.7)	0.2795	0.0901	0.5826
**Male gender**	39 (93)	17 (94.4)	10 (83.3)	12 (100)	0.5478	1.0000	0.4783
**Prehospital clinical condition**							
**Initial signs of life**	35 (83.3)	13 (72.2)	11 (91.7)	11 (100)	1.0000	1.0000	1.0000
**Spontaneous ventilation**	14 (60.8)	NA	5 (46.7)	9 (100)	NA	NA	0.0071 *
**Limb movement**	11 (26.2)	NA	3 (50)	9 (100)	NA	NA	0.0011 *
**Heart rate**	120 (12.5–140)	NA	85 (0–147)	120 (110–140)	NA	NA	0.5207
**Hypotension**	9 (64.3)	NA	6 (85.7)	3 (42.8)	NA	NA	0.094
**Glasgow score**	14 (3–15)	NA	3 (3–11)	15 (15–15)	NA	NA	0.0038 *
**Prehospital management**							
**Attempted resuscitation**	35 (83,3)	14 (93.3)	12 (100)	9 (100)	1.000	1.0000	1.0000
**Intubation**	11 (42.3)	1 (33.3)	8 (66.7)	2 (20)	0.5253	1.0000	0.0361 *
**Transfusion**	2 (0.6)	1 (7.7)	1 (8.3)	0 (0)	1.000	1.0000	1.0000
**Chest drain placement**	11 (42.3)	4 (30.8)	4 (36.4)	0 (0)	1.0000	0.0983	0.0902
**Cardiorespiratory arrest**	13 (44.8)	5 (100)	8 (66.7)	0 (0)	0.2605	0.0002 *	0.0013 *
	**All patients**	**Group A**	**Group B**	**Group C**	** *p* ** **-value A–B**	** *p* ** **-value A–C**	** *p* ** **-value B–C**
**External cardiac massage**	9 (40.9)	2 (66.7)	7 (87.5)	0 (0)	0.4909	0.0330 *	0.0002 *
**Catecholamines**	11 (45.6)	1 (50)	8 (72.7)	2 (18.2)	1.0000	0.4231	0.0300 *
**Adrenaline**	9 (37.5)	1 (100)	7 (63.6)	1 (9.1)	1.0000	0.2949	0.0237 *
**Noradrenaline**	4 (16.7)	0 (0)	3 (27.3)	1 (9.1)	1.0000	1.0000	0.5865
**Volume expansion >1 L**	5 (31.2)	1 (50)	3 (50)	1 (12.5)	1.000	0.3778	0.2448
**Clinical status at hospital arrival**							
**care time (call to arrival at the hospital)**		55 (49.5–69)	52 (33.5–63)	62.5 (54–79.5)	0.368	0.313	0.028 *
**Dead on arrival**	2 (4.8)	NA	2 (20)	0 (0)	NA	NA	0.016 *
**Tachycardia**	13 (59)	NA	5 (50)	8 (66.7)	NA	NA	0.6656
**Hypotension**	16 (72)	NA	9 (90)	7 (58.3)	NA	NA	0.1619
**Glasgow score**	14 (3–15)	NA	3 (3–7)	15 (14–15)	NA	NA	0.0024 *
**State of shock on arrival**	14 (58)	NA	12 (100)	2 (16.7)	NA	NA	0.0128 *
**Cardiorespiratory arrest**	9 (37.5)	NA	9 (75)	0 (0)	NA	NA	0.0003 *
**In-hospital management**							
**Preoperative mechanical ventilation**	11 (50)	NA	10 (90.1)	1 (9.1)	NA	NA	0.0003 *
**Preoperative volume expansion >1 L**	17 (74)	NA	10 (90.1)	7 (58.3)	NA	NA	0.1550
**Preoperative catecholamines**	14 (58)	NA	12 (100)	2 (16.7)	NA	NA	<0.0001 *
**Adrenaline**	11 (46)	NA	11 (91.7)	0 (0)	NA	NA	<0.0001 *
**Noradrenaline**	6 (25)	NA	4 (33.3)	2 (16.7)	NA	NA	0.6404
**Lowest haemoglobin at arrival (g/dL)**	9.7 (8–10.5)	NA	9.4 (7.9–10)	9.9 (8.2–12.4)	NA	NA	0.1096
**Preoperative transfusion**	5 (21)	NA	4 (33.3)	1 (8.3)	NA	NA	0.3168
**Preoperative chest tube placement**	7 (27)	NA	5 (41.7)	2 (16.7)	NA	NA	0.3707
	**All patients**	**Group A**	**Group B**	**Group C**	** *p* ** **-value A–B**	** *p* ** **-value A–C**	** *p* ** **-value B–C**
	**Complementary investigations**						
**FAST ultrasound**	18 (75)	NA	7 (58.3)	11 (91.7)	NA	NA	0.1550
**Pericardial effusion**	14 (78)	NA	5 (41.7)	9 (75)	NA	NA	1.0000
**Tamponade**	6 (30)	NA	2 (16.7)	4 (33.3)	NA	NA	1.0000
**Preoperative thoracic CT**	4 (16)	NA	1 (8.3)	3 (25)	NA	NA	0.5901
**Pneumothorax on CT**	4 (16)	NA	1(8.3)	1 (8.3)	NA	NA	1.0000
**Haemothorax on CT**	13 (60)	NA	1 (8.3)	2 (16.7)	NA	NA	1.0000
	**Surgery**						
**Surgery performed**	22 (52)	NA	10 (83.3)	12 (100)	NA	NA	0.2174
**Salvage thoracotomy**	4 (17)	NA	3 (25)	1 (8.3)	NA	NA	0.5901
**Sternotomy**	16 (38)	NA	8 (66.7)	8 (66.7)	NA	NA	1.0000
**Thoracotomy**	9 (18)	NA	3 (25)	5 (41.7)	NA	NA	0.6668
**Death in OR**	7 (17)	NA	7 (70)	0 (0)	NA	NA	0.0007 *
**Time from hospital arrival to OR (min)**	25 (15–55)	NA	15 (15–25)	97 (74–121)	NA	NA	0.1000
**Time in OR (min)**	98 (60–140)	NA	62 (57–100)	105 (86–132)	NA	NA	0.1264
**Perioperative blood**	500 (0–1200)	NA	500 (0–1050)	500 (250–1650)	NA	NA	0.6335
**Perioperative cardiac death**	7 (16.7)	NA	7 (58.3)	0 (0)	NA	NA	0.0274 *
**Perioperative catecholamines**	16 (72.7)	NA	10 (100)	6 (50)	NA	NA	0.0152 *
**Adrenaline**	9 (40.9)	NA	9 (90)	0 (0)	NA	NA	<0.0001 *
**Noradrenaline**	11 (50)	NA	5 (50)	6 (50)	NA	NA	1.0000
**Number of units transfusion**	4 (0.5–6.5)	NA	4 (4–8)	2 (0–4)	NA	NA	0.0394 *
**Perioperative FFP**	13 (61.9)	NA	7 (77.8)	6 (50)	NA	NA	0.3383
**Number of units**	2 (0–4)	NA	3 (1–4)	0.5 (0–4)	NA	NA	0.2708
**Lowest haemoglobin level at day 1 (g/dL)**	9.5 (7.6–12)	NA	7.3 (3.7–7.3)	11.7 (9.4–12.0)	NA	NA	0.0026 *
**Early postoperative death**	1 (2.4)	NA	1 (8.3)	NA	NA	NA	NA
	**All patients**	**Group A**	**Group B**	**Group C**	** *p* ** **-value A–B**	** *p* ** **-value A–C**	** *p* ** **-value B–C**
**Late postoperative death**	2 (4.8)	NA	2 (16.7)	NA	NA	NA	NA
**Postoperative Adrenaline**	4 (25)	NA	3 (25)	0 (0)	NA	NA	0.0005 *
**Postoperative noradrenaline**	7 (43.7)	NA	3 (25)	4 (33.3)	NA	NA	0.2615
**Postoperative catecholamines for 24 h**	3 (7.9)	NA	1 (8.3)	2 (16.7)	NA	NA	1.0000
**Duration of hospitalisation in resuscitation and intensive care (days)**	2 (0–5.2)	NA	0 (0–0.5)	3.5 (2–5.2)	NA	NA	0.0207 *
**Duration of hospitalisation excluding resuscitation and intensive care (days)**	4 (0–6.75)	NA	0 (0–0)	5.5 (4–9.7)	NA	NA	0.0001 *
**Total duration of hospitalisation (days)**	7 (0–11)	NA	0 (0–0.5)	9 (7–14.2)	NA	NA	0.0013 *

Data are expressed as medians (25th–75th percentiles) and quantitative data as numbers (percentage). NA: insufficient or unreported data or variable not applicable in this population; OR: operating room; CT: computed tomography. Percentages are expressed in relation to the amount of data available for each column. * mean statistically significant.

**Table 2 diagnostics-14-01406-t002:** Variables studied and relation to mortality.

Variables	Mortality (%)	Odd Ratio	95% CI	*p*
**Prehospital management**				
Absence of limb movement	100	4	[1.50–10.66]	0.0003
Absence of spontaneous ventilation	100	2.8	[1.39–5.65]	0.0026
Prehospital drain placement	100	1.7	[1.23–2.31]	0.0370
Prehospital cardiorespiratory arrest	100	4	[1.71–9.35]	<0.0001
Prehospital external cardiac massage	100	6.5	[1.82–23.26]	0.0002
Prehospital catecholamines	81.8	2.6	[1.12–6.29]	0.0188
Prehospital adrenaline	88.9	2.7	[1.26–5.66]	0.0131
**In-hospital management**				
Preoperative shock	81.8	2.6	[1.12–6.29]	0.0188
In-hospital cardiorespiratory arrest	100	4	[1.71–9.35]	0.0013
Preoperative mechanical ventilation	90.9	10	[1.53–65.41]	0.0003
Preoperative catecholamines	85.7	7	[1.94–25.25]	<0.0001
Preoperative adrenaline	100	13	[1.98–85.46]	<0.0001
Preoperative cardiorespiratory arrest	100	5	[1.82–13.76]	0.0003
Sternotomy	50	0.6	[0.35–0.99]	0.0324
Thoracotomy	37.5	0.5	[0.19–1.17]	0.0309
Peroperative catecholamines	62.5	2.7	[1.42–5.02]	0.0152
Peroperative adrenaline	100	13	[1.98–85.46]	<0.0001
**Injury characteristics**				
Injury in attack and defence areas	86.9	1.6	[1.05–2.60]	0.0195
Single cardiac injury	58.3	0.6	[0.45–0.95]	0.0415
Multiple cardiac injuries	88.9	1.5	[1.05–2.22]	0.0415
Penetrating cardiac injury	84.8	14	[2.10–93.22]	0.0064 *
AAST-OIS score 5	89.5	2.9	[1.04–8.54]	0.0329

*p* = *p* value; Mortality: number of deceased patients with the variable present divided by the total number of patients with the variable present. * mean statistically significant.

## Data Availability

Data is contained within the article or Appendix A.

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
