# Peer review of "Resuscitation and Forensic Factors Influencing Outcome in Penetrating Cardiac Injury"

_diagnostics, 2024, doi:10.3390/diagnostics14131406_

Round 1

Reviewer 1 Report

Comments and Suggestions for Authors

I thank the authors for the opportunity they have given me to read this interesting manuscript.

The authors calculate various scales (such as AAST score). In the methodology they must reference each scale and explain how it is calculated (even if it is with tables in supplementary material).

In table 1 I find multiple errors in the calculation of the percentages (e.g. "spontaneous ventilation" 14/42 indicate that it is 60.8% - it is 33.33% - and 9/12 is not 100% and so on in multiple data throughout the entire table). Authors must review the entire table and correct errors, as well as their transcription into the text.

When the authors present the AUC of the AAST score with its confidence interval, it would be interesting to also know the "p-value". Furthermore, for the cut-off point "5" selected by the authors, sensitivity and specificity should be provided with their confidence intervals, positive and negative predictive values, LR+ and LR-.

Author Response

Rewiewing 1 :

Comments 1 : The authors calculate various scales (such as AAST score). In the methodology they must reference each scale and explain how it is calculated (even if it is with tables in supplementary material).

Response 1 : Thank you for pointing this out. We have added this data to the appendix section 4.

Comments 2 : In table 1 I find multiple errors in the calculation of the percentages (e.g. "spontaneous ventilation" 14/42 indicate that it is 60.8% - it is 33.33% - and 9/12 is not 100% and so on in multiple data throughout the entire table). Authors must review the entire table and correct errors, as well as their transcription into the text.

Response 2 : Thank you for pointing this out. The discrepancy arises from the fact that the percentages were calculated by subtracting the missing data. We add  « Percentages are expressed in relation to the number of data available for each column. »

Comments 3 : When the authors present the AUC of the AAST score with its confidence interval, it would be interesting to also know the "p-value". Furthermore, for the cut-off point "5" selected by the authors, sensitivity and specificity should be provided with their confidence intervals, positive and negative predictive values, LR+ and LR-.

Response 3 : Thank you for pointing this out. We add « with a specificity of 80.0%, a sensibility of 56.6%, a positive likelihood ratio of 2.83 and a negative likelihood ratio of 0.54 »

Rewiewing 2/

Comments 1 : I read this paper with great interest and I think the results are well presented. However, I would like to point out the possible presence of an error in lines 165-166: it seems that the percentage values have been reversed.

Response 1 : Thank you for pointing this out, we correcte it.

Rewiewing 3/

Comments 1 : Thank you very much for inviting me to review this article. This is a very interesting manuscript regarding the management of victims of penetrating cardiac injuries before and after admission to hospital and the anatomical characteristics of these injuries in order to facilitate the diagnosis of the most critical patients. I believe that this work is written very well - the introduction is comprehensive, the methodology and results are consistent, the discussion and conclusions are written as correctly as possible and I have no objections to them. One thing that the authors need to improve is the expansion of abbreviations in the text (especially the abstract) - because even though everyone in this field knows what it means - the professional literature requires it (e.g. AAST-OIS score or AUC-ROC). Apart from that, I really congratulate the authors on their very good work and I appreciate the hard effort they put into researching and preparing this manuscript.

Response 1 : We thank you for your encouragement and your review. We improve the expension of abbreviation.

Reviewer 2 Report

Comments and Suggestions for Authors

I read this paper with great interest and I think the results are well presented. However, I would like to point out the possible presence of an error in lines 165-166: it seems that the percentage values have been reversed.

Author Response

Rewiewing 2/

Comments 1 : I read this paper with great interest and I think the results are well presented. However, I would like to point out the possible presence of an error in lines 165-166: it seems that the percentage values have been reversed.

Response 1 : Thank you for pointing this out, we correcte it.

Reviewer 3 Report

Comments and Suggestions for Authors

Thank you very much for inviting me to review this article. This is a very interesting manuscript regarding the management of victims of penetrating cardiac injuries before and after admission to hospital and the anatomical characteristics of these injuries in order to facilitate the diagnosis of the most critical patients. I believe that this work is written very well - the introduction is comprehensive, the methodology and results are consistent, the discussion and conclusions are written as correctly as possible and I have no objections to them. One thing that the authors need to improve is the expansion of abbreviations in the text (especially the abstract) - because even though everyone in this field knows what it means - the professional literature requires it (e.g. AAST-OIS score or AUC-ROC). Apart from that, I really congratulate the authors on their very good work and I appreciate the hard effort they put into researching and preparing this manuscript.

Author Response

Rewiewing 3/

Comments 1 : Thank you very much for inviting me to review this article. This is a very interesting manuscript regarding the management of victims of penetrating cardiac injuries before and after admission to hospital and the anatomical characteristics of these injuries in order to facilitate the diagnosis of the most critical patients. I believe that this work is written very well - the introduction is comprehensive, the methodology and results are consistent, the discussion and conclusions are written as correctly as possible and I have no objections to them. One thing that the authors need to improve is the expansion of abbreviations in the text (especially the abstract) - because even though everyone in this field knows what it means - the professional literature requires it (e.g. AAST-OIS score or AUC-ROC). Apart from that, I really congratulate the authors on their very good work and I appreciate the hard effort they put into researching and preparing this manuscript.

Response 1 : We thank you for your encouragement and your review. We improve the expension of abbreviation.